# Amitraz Resistance in French *Varroa* Mite Populations—More Complex Than a Single-Nucleotide Polymorphism

**DOI:** 10.3390/insects15060390

**Published:** 2024-05-27

**Authors:** Ulrike Marsky, Bénédicte Rognon, Alexandre Douablin, Alain Viry, Miguel Angel Rodríguez Ramos, Abderrahim Hammaidi

**Affiliations:** 1Véto-Pharma, 12 Rue de la Croix Martre, 91120 Palaiseau, France; ulrike.marsky@vetopharma.com; 2Biomnigene, 18 Rue Alain Savary, 25000 Besançon, France; 3LDA39, Laboratoire Départemental d’Analyses du Jura, 59 Rue du Vieil Hôpital, 39800 Poligny, France

**Keywords:** *Varroa* mite, resistance, variation, single-nucleotide polymorphism, honey bee, amitraz

## Abstract

**Simple Summary:**

Our study addresses the growing number of reports of *Varroa* mite resistance to amitraz, which is vital for beekeepers in managing this destructive pest. Recently, beekee-pers in France and the United States have reported the diminishing effectiveness of amitraz treatments, threatening their ability to control *Varroa* infestations. We collected data from French apiaries in 2020 and 2021, examining the field effectiveness of amitraz treatments and conducting laboratory sensitivity tests to the pesticide. Additionally, we genotyped 240 *Varroa* mites from different regions in France, focusing on a specific genetic marker previously linked to amitraz resistance. Our findings revealed regional occurrences of resistance, but the decrease in treatment efficacy was less severe compared to resistance against other pesticides. Notably, our genetic analysis suggested that the previously identified genetic marker may not directly cause amitraz resistance. These findings underscore the urgent need for ongoing monitoring and the development of new strategies to manage *Varroa* mites, ensuring the sustainability of beekeeping. By understanding and addressing resistance, we can maintain amitraz as a reliable tool to protect bee colonies, which are crucial due to the essential pollination services they provide.

**Abstract:**

Resistance against amitraz in *Varroa* mite populations has become a subject of interest in recent years due to the increasing reports of the reduced field efficacy of amitraz treatments, especially from some beekeepers in France and the United States. The loss of amitraz as a reliable tool to effectively reduce *Varroa* mite infestation in the field could severely worsen the position of beekeepers in the fight to keep *Varroa* infestation rates in their colonies at low levels. In this publication, we present data from French apiaries, collected in the years 2020 and 2021. These data include the field efficacy of an authorized amitraz-based *Varroa* treatment (Apivar^®^ ,Véto-pharma, France) and the results of laboratory sensitivity assays of *Varroa* mites exposed to the reference LC_90_ concentration of amitraz. In addition, a total of 240 *Varroa* mites from Eastern, Central, and Southern regions in France that were previously classified as either “sensitive” or “resistant” to amitraz in a laboratory sensitivity assay were genotyped. The genetic analyses of mite samples are focused on the β-adrenergic-like octopamine receptor, which is considered as the main target site for amitraz in *Varroa* mites. Special attention was paid to a single-nucleotide polymorphism (SNP) at position 260 of the *ORβ-2R-L* gene that was previously associated to amitraz resistance in French *Varroa* mites, *Varroa*. Our findings confirm that amitraz resistance occurs in patches or “islands of resistance”, with a less severe reduction in treatment efficacy compared to pyrethroid resistance or coumaphos resistance in *Varroa* mites. The results of our genetic analyses of *Varroa* mites call into question the hypothesis of the SNP at position 260 of the *ORβ-2R-L* gene being directly responsible for amitraz resistance development.

## 1. Introduction

The efficient and sustainable control of *Varroa* mites (*Varroa destructor*) in honey bee colonies (*Apis mellifera*) is characterized by ongoing challenges for beekeepers at a global scale [1]. Although several *Varroa* treatments with different active ingredients have been available for several decades, high colony losses are periodically reported by beekeepers from different global regions [2,3,4]. Multiple factors, such as environmental changes (climate and human land use changes) [5], inter-seasonal variations in weather, the available nectar and pollen sources, as well as honey bee and *Varroa* population growth [6,7], and the development of acaricide resistance in *Varroa* mites [8], can contribute to high colony losses due to varroosis. In addition, the complete lack of control options for bee viruses, which are transmitted by *Varroa* mites, allows for ever-increasing viral loads in honey bee colonies that build up continuously over time [9]. 

In beekeeping, one of the most prominent examples of acaricide resistance development in *Varroa* mites is pyrethroid resistance [10]. These molecules have been primarily formulated and applied in honey bee colonies as long-term treatments (over a period of several weeks) in the form of plastic strips. Tau-fluvalinate resistance appeared in Lombardy, Italy, soon after its 1989 authorization [11], and in the United States after its 1990 approval [12], leading to severely decreased treatment efficacy by the late 1990s [13]. *Varroa* resistance against flumethrin, another pyrethroid that has been authorized for *Varroa* control in countries all over the world, has been detected repeatedly over the last few decades—sometimes as cross-resistance against both tau-fluvalinate and flumethrin [14,15,16]. Once it occurred, the resistance of *Varroa* mites to pyrethroids spread quickly and led to severe reductions in treatment efficacy [10,11]. 

The underlying genetic mechanisms leading to pyrethroid resistance in *Varroa* mites have been investigated extensively over the years [10,17,18,19]. Mite populations from di-fferent countries were genotyped to identify polymorphisms that could provide more information on possible resistance mechanisms at the phenotypical level. In several research studies, single-nucleotide polymorphisms (SNPs) were identified as the source of tau-fluvalinate and/or flumethrin resistance in *Varroa* mites. Although the location of SNPs in the mites’ genome varied geographically, the main target mechanism of the detected variations appeared to be the same: an alteration in the voltage-gated sodium channels, the main target site for pyrethroids [20,21,22].

For many years, amitraz (a formamidine) withstood any signs of resistance development in most regions where it was used for *Varroa* control. The majority of colonies (62%) in Europe receive treatment with amitraz, either through strips or fumigation. In eight Western European countries (cluster I), amitraz strips are predominantly utilized, whereas in seven Eastern European countries (cluster III), amitraz fumigation is the do- minant method [23]. In Poland, amitraz use in beekeeping dates to the beginning of the 1980s. Despite a now 40+ year treatment history of amitraz in the country’s bee hives, it is still applied at present, resulting in successful treatment outcomes [24]. Similarly, in Spain, amitraz-based *Varroa* control products date back to the late 1990s. To date, the molecule remains an effective component of *Varroa* control in Spanish beekeeping operations [25].

In recent years, data from the United States and France have raised concerns that amitraz resistance among *Varroa* mite populations in these countries appears to be deve- loping in more and more apiaries [26,27]. In the United States, many amitraz resistance reports are based on field bioassay data (also referred to as “Pettis’s test” in beekeeping, after Dr. Jeff Pettis) [27,28,29]. In France, laboratory sensitivity assays, detecting the susceptibility of *Varroa* mite populations to amitraz, have been utilized as the main tool to identify amitraz resistance in French *Varroa* mite populations [26,28,30]. Interestingly, the occurrence of amitraz resistance in apiaries across both countries has been characterized very differently to pyrethroid resistance. Instead of a resistant *Varroa* mite phenotype that spreads quickly within and across apiaries and leads to severe reductions in field efficacy, amitraz resistance appears to manifest in “patches” or “islands of resistance” with—at least initially—a less severe reduction in field efficacy within the affected apiaries [27].

Concerning the genetic basis of *Varroa* mite resistance to amitraz, similarly to pyrethroid resistance, single-nucleotide polymorphisms have been proposed as a potential cause of amitraz resistance in French and American *Varroa* mite populations. By comparing the sequences of octopamine and tyramine receptor (the known targets of amitraz) genes obtained from mites collected in different apiaries with different treatment regimens in France and the United States, Hernández Rodríguez et al. indeed identified two variations in the β-adrenergic-like octopamine receptor gene *Orβ-2R-L* potentially associated with amitraz treatment failure events [28]. The first variation, located at position 260 in the *Orβ-2R-L* gene (substitution of A with G at nucleotide 260 within the ORF), was exclusively detected in French mites. The resulting amino acid substitution within the β-adrenergic-like octopamine receptor, N87S (asparagine (AAT) to serine (AGT) substitution at position 87), is positioned at the end of helix II near the residues thought to constitute the binding site for octopamine. This variation is likely to disturb the correct folding of the protein and, therefore, the efficient binding of the octopamine and of its antagonist, the amitraz. The second variation identified (substitution of T with C at position 643 of the ORF) was detected only in the mites collected in the U.S. It resulted in a tyrosine to histidine substitution at position 215 (Y215H) within the octopamine receptor due to the transition in the corresponding code (TAT to CAT). Located in the fifth transmembrane segment of the protein, this substitution is predicted to strongly reduce the stability of the receptor and to affect the interaction with its ligand.

The topic of the development of resistance against active ingredients of acaricides in *Varroa* mite populations has received increasing attention in recent years [20,21,22,25]. There are increasing concerns among beekeepers, veterinarians, bee inspectors, and honey bee health researchers due to an increasing number of reports of amitraz resistance in *Varroa* mite populations from the field. Most of these reports stem from beekeeping operations in the United States and France, with ongoing investigations in Canadian beekeeping operations underway [26,27,31]. 

The objective of the present study was twofold: 

Gain information about the spread and magnitude of amitraz resistance in the field in French apiaries. 

Investigate genetic mechanisms underlying amitraz resistance in French *Varroa* mite populations, focusing on the single-nucleotide polymorphism at position 260 in the *Orβ-2R-L* gene described by Hernández Rodríguez et al. (2021) given their identification in French *Varroa* populations [28].

To achieve this, the following activities were carried out during the years 2020 and 2021: (1) assessment of the treatment efficacy of an authorized *Varroa* control product (Apivar^®^) in the field; (2) assessment of the laboratory sensitivity of *Varroa* mites to amitraz; (3) collection and analysis of genotyping data from French *Varroa* mite populations.

Our study focuses on variation rather than mutation, using the term “variant” to describe alterations that can be benign, pathogenic, or of unknown significance. “Variant” is increasingly replacing “mutation” in scientific discourse.

## 2. Materials and Methods

### 2.1. Organization of the Study

In 2020 and 2021, Véto-pharma conducted laboratory assays to detect *Varroa* mite susceptibility to amitraz, which were carried out by the Laboratoire Départemental d’Analyses du Jura (LDA39). *Varroa* mites were sampled from worker and drone brood frames collected from different commercial apiaries across France and sent to the LDA39. *Varroa* mites were then exposed to a predetermined concentration of amitraz (LC_90_: Lethal Concentration for 90% of the study population) in laboratory tests. The sensitivity to amitraz of *Varroa* populations from different regions of France was assessed by measuring the mortality rate of the mites during the assay as an indicator of susceptibility. The 2021 assays aimed to consolidate the initial results obtained by both refining the LC_90_ determination and conducting additional sensitivity tests with *Varroa* samples from 10 different beekeepers.

*Varroa* mites, selected from both the dead and surviving mites tested in the laboratory assays, were later genotyped by Biomnigene laboratory. In 2021, laboratory susceptibility tests were paired with field efficacy tests using the authorized *Varroa* treatment, Apivar^®^. These field tests were carried out by the French commercial beekeepers’ Association pour le Développement de l’Apiculture provençale (ADAPI) in Provence-Alpes-Côte d’Azur (PACA) and Association pour le Développement de l’Apiculture in Occitanie (ADA Occitanie).

### 2.2. Apiary Locations and Samples’ Origin

To determine the reference LC_90_, *Varroa* populations were gathered in May 2020 from two “organic” apiaries from Bretagne and Occitanie, and in May 2021 from two additional “organic” apiaries in Occitanie and PACA, where amitraz treatments had not been used. Samples were collected from five to six colonies within each apiary. For this preliminary investigation, *Varroa* mites were pooled by apiary, regardless of colony origin. 

For the main susceptibility assays, *Varroa* mite samples were collected in June and July 2020 from 12 apiaries across 6 French regions: Nouvelle-Aquitaine, Occitanie, PACA, Auvergne-Rhône-Alpes, Centre-Val de Loire, and Grand Est. Acknowledging the inherent variability in mite sensitivity among hives within the same apiary, sensitivity evaluations were conducted at the colony level. A sample consisted of *Varroa* mites from one or several frames of capped brood from a single colony. Two or three colonies were randomly sampled in each apiary.

In 2021, our focus was on two key honey-producing regions in the South of France: Occitanie and PACA. *Varroa* mite samples were collected from 10 apiaries in these regions. In each selected apiary, mite samples were systematically gathered from six brood frames across six individual colonies. Sampling occurred over a period of five weeks, from 7 June to 6 July, with two apiaries examined per week. Given the unexpectedly low *Varroa* infestation observed during the initial phase, supplementary sampling sessions were conducted between 19 July and 13 September.

The field efficacy trial with Apivar^®^ was carried out on two apiaries with 25 colonies each, located in Auzeville-Tolosane (Department 31, Occitanie) and Avignon (Department 84, PACA region). The hives of the five beekeepers from each region were moved to these two apiaries at the end of August 2021. Apivar^®^ treatment was inserted on 31 August in Auzeville and on 14 September in Avignon.

## 3. Method for Laboratory Susceptibility Test

### 3.1. Principle of the Method

As amitraz exhibits lipophilic properties and the amitraz-based treatments used in France are based on contact action with impregnated strips [23], the methodology employed in this study to set up laboratory susceptibility tests of *Varroa* mites originates from a standardized procedure outlined in the Coloss Beebook, specifically tailored for fat-soluble substances acting through contact (§3.6.3.2 of the Coloss Beebook [32]). This approach draws upon the work laid by Milani [33]. The acaricide under investigation was integrated into paraffin capsules.

### 3.2. Preparation of Paraffin Capsules Coated with Amitraz

These paraffin capsules were prepared as described in Milani (1995) [33] and in *La Santé de l’Abeille* [34]. They were formed using two Na-Ca glass discs (62 mm in diameter) and one stainless steel ring (56 mm inside diameter; 2 mm thick; 5 mm high). The interior of these capsules is entirely covered by a thin layer of paraffin (Merck ref. 1.07151.1000—melting point 46–48 °C) containing a known concentration of amitraz (Amitraz Pestanal^®^ Sigma-Aldrich, MA, USA—ref. 45323-250MG). For each preparation, 10 g of paraffin was melted in a crystallizer placed in a water bath at 60 °C, to which the required quantity of amitraz was added and dissolved in 2 mL hexane (n-Hexane; VWR; ref. 24577.323). For negative controls, hexane alone was added.

### 3.3. Varroa Sampling and Laboratory Exposure of Varroa Mites to Amitraz

*Varroa* mites were harvested within 24 h of receiving the frames or brood fragments by individually uncapping each cell. Only mature *Varroa* females (founder females or young mature females) extracted from capped cells were utilized for the assay. These were brought into contact with the treated paraffin. The mites’ contact time with the miticide was shortened to 1 h instead of the 4 h described in the Beebook method. Previous LDA39 trials, not described here, showed that LC_90_ for amitraz and for this method was at much lower levels compared to the research on pyrethroids by Milani [33]. To avoid possible biases related to the preparation of capsules at very low concentrations, the contact time of 1 h was considered more appropriate to achieve LC_90_ levels between 10 and 30 ppm, similar to those obtained for pyrethroids. 

Following the incubation period, mites were transferred to a Petri dish containing two or three new larvae/nymphs of worker bees. Capsules and Petri dishes were incubated at 32.5 °C with a relative humidity (RH) of approximately 70%. Mortality rates were then counted 24 h later.

During observation under a binocular magnifying glass, the mites were classified into three distinct categories:-Mobile mites: mites characterized by spontaneous movement or movement in response to external stimuli, such as brushing.-Paralyzed mites: mites exhibiting the ability to move one or more appendages but an inability to relocate.-Dead mites: mites showing no observable response following three consecutive stimuli.

Assessment was conducted upon completion of exposure to the acaricide, during the transfer of mites from the capsules to Petri dishes. Mites that were inadvertently lost or fatally injured, for instance, those entrapped between the metal ring and a glass disc, were excluded from the analysis. This assessment was reiterated 24 h post-introduction into the capsules, denoted as T0 + 24. The action of amitraz, which targets the mites’ octopamine receptors at the level of synaptic transmission and which leads to paralysis of the mites, is sub-lethal [35,36]. The death of *Varroa* mites occurs secondarily due to starvation related to paralysis. Therefore, mites that were paralyzed after 24 h, unable to move, were counted in the same category as dead mites.

### 3.4. Determination of Reference LC_90_ Concentration of Amitraz

As this bioassay method for substances’ contact with impregned paraffin capsules had never been used on amitraz, to our knowledge, prior to conducting laboratory sensitivity tests, it was imperative to establish the reference LC_90_) for amitraz in susceptible *Varroa* mite populations. To achieve this, *Varroa* populations from “organic” apiaries were selected for sampling. The selected apiaries for determining the reference LC_90_ were chosen based on the following criteria: -A minimum interval of 5 years devoid of amitraz application was required, in a-ccordance with observations indicating a reversion of *Varroa* resistance in populations unexposed to pyrethroids for at least 4 years [37,38].-Recent introduction of new queens or merging with colonies exposed to amitraz within the previous 5 years was to be avoided.-Absence of prevalent diseases such as American or European foulbrood, or nosemosis, was obligatory.-Satisfactory nutritional provisions and the inclusion of renewed frames were prere-quisites.

A series of twelve concentrations, including a negative control group, underwent testing according to the following scheme: -Three replicates consisting of 10 *Varroa* mites per apiary were subjected to concentrations of 0, 0.5, 1, 2, 3, 5, 7.5, and 10 ppm.-Two replicates, each comprising 10 *Varroa* mites per apiary, were exposed to a concentration of 12.5 ppm.-A single replicate containing 10 *Varroa* mites per apiary was administered to concentrations of 15, 20, 50, and 100 ppm.

To enhance the resolution of data surrounding the dose eliciting a 90% mortality rate in 2020, adjustments were made to the dilution range in 2021, as follows: -Three replicates were conducted at the following concentrations: 0, 1, 5, 7.5, 10, 12.5, 15, 20, 25, and 30 ppm.-A single replicate was administered to concentrations of 40, 50, and 100 ppm.

When there were not enough *Varroa* mites in the brood, the tests were adapted and carried out with 8 *Varroa* mites instead of 10 per capsule replicate.

### 3.5. Laboratory Sensitivity of Varroa Mites to the LC_90_

For each sample, five replicates with 8 or 10 mites per replicate were exposed to the baseline LC_90_ for 1 h, resulting from the preliminary work. In addition, three replicates containing 8 or 10 *Varroa* mites served as negative controls.

### 3.6. Data Analysis

The percentage of miticide effect (mortality/paralysis) was calculated as follows:



The percentage of effect obtained for the different concentrations was corrected to subtract the “natural mortality” according to the Schneider–Orelli formula: (b − k)/(1 − k) with b denoting the percentage of effect in the test and k denoting the percentage of effect in the negative control [34]. 

LC_90_ calculations were obtained using the probit transformation method. The confidence interval of the LC_90_ was established with a 5% risk of error. The coefficient of determination (R^2^) indicates the accuracy of the linear regression predictions. An R^2^ value close to 1 signifies a strong prediction, while a value near 0 indicates a weak prediction. The correlation coefficient (r) measures the strength of the correlation between the acaricide concentration and the percentage of effect variables. A correlation is considered strong when (r) is between 0.5 and 1, and weak when it is between 0 and 0.5.

The trials for which the percentage of effect in the negative controls was greater than 30% were invalidated according to the Beebook method (cf.§3.6.3.2 in Beebook II [32]). 

*Varroa* mite populations were categorized into three groups according to the percentage of effect obtained at the LC_90_: -Sensitive population group: *Varroa* mites that achieved a percentage of effect at a LC_90_ between 75 and 100%.-Intermediate population group: *Varroa* mites that achieved a percentage of effect at a LC_90_ between 60 and 74%.-Resistant population group: *Varroa* mites that achieved a percentage of effect at a LC_90_ under 60%.

This classification was based on results from similar studies [1,4,6,8,13].

## 4. Genotyping of the *ORβ-2R-L* Gene and Potential Links to Amitraz Resistance

### 4.1. Genomic DNA Extraction

Genomic DNA extraction from the mites to be genotyped was carried out as described by González-Cabrera et al. (2013) [17]. Briefly, individualized mites transferred in 1.5 Eppendorf tubes were first washed with 1 mL of sterile water to remove the traces of ethanol in which they were kept until analysis. After water was removed, they were incubated at 99 °C for 10 min in 20 µL of 0.25 M NaOH. They were subsequently crushed using a plastic homogenizer and the solution was neutralized adding 10 µL of 0.25 M HCl, 5 µL of 0.5 M Tris-HCl, and 5 µL of 2% Triton X-100 before a last incubation at 99 °C for 10 min. Genomic DNA suspensions were stored at −20 °C.

### 4.2. Genotyping by Sanger Sequencing

Amplification of the *ORβ-2R-L* gene by PCR was performed in a reaction mixture of 25 µL containing 1.5 µL of genomic DNA, 12.5 µL of AccuStart II PCR SuperMix 2X (Quantabio, MA, USA) and 0.2 M of each primer. The PCR program consisted of an initial denaturation step at 95 °C for 10 min, followed by 40 cycles of 95 °C for 20 s, 55 °C for 20 s, and 70 °C for 1 min, and a final elongation step at 70 °C for 1 min. The primers used were ORβ-2R-L_Seq For (5′-GCTGATCTCGATCATATTGAC-3′) and ORβ-2R-L_Seq Rev (5′-CTCGAGTGGCTTGATGATCGC-3′), which were designed with the program Primer 3 u-sing a sequence of the *ORβ-2R-L* gene available in the NCBI database (accession number: XM_022808962.1) as a reference. They allow the amplification of a short fragment of the gene (304 bp) covering the targeted variation at position 260. After checking their length with capillary electrophoresis, PCR products were purified using 2 µL of ExoSAP IT^TM^ (Applied Biosystems) for 5 µL of gDNA. PCR Sanger reactions were performed using 3 µL of purified fragments and the primers that were used for the amplification step. Sanger PCR products were run using SeqStudio (Applied Biosystems) and the obtained sequences were aligned with the reference sequence of the gene (accession number: XM_022808962.1). 

## 5. Treatment Efficacy of Apivar^®^ in the Field

### Treatment Efficacy of Apivar^®^—Protocol 2021 (PACA and Occitanie Region)

The installation of the Apivar^®^ treatment was carried out on August 31 in Auzeville (Occitanie region) and on 14 September in Avignon (PACA region). 

Each hive was equipped with a screened bottom-board, allowing for *Varroa* mite counts to easily be obtained on white sheets of sticky paper. Counts were performed twice a week right after the application of treatments and once a week later on. Initial counts were conducted from 11 to 14 days before the application of the Apivar^®^ treatment. Mite counts were then continued until three weeks after the application of the second control treatment (see below: control treatments).

On the day of application of the Apivar^®^ strips (Day 0 or D0) and on the day of their removal (Day 70 or D70), the colonies were evaluated with the ColEval method [39], weighed, and sampled to assess the rate of phoretic *Varroa* mites per 100 bees (PVM100bees). On day 35 of the treatment, the strips were cleaned with a hive tool and repositioned in the center of the brood nest (Figure 1).

Two control treatments were applied. When the Apivar^®^ (D70) strips were removed, the queens were caged and four Bayvarol^®^ (flumethrin) strips were inserted between the brood frames. After 28 days, these strips were removed, the queens released, and an Oxybee^®^ (oxalic acid) treatment was applied at a rate of 5 mL per occupied interframe. 

The average treatment efficacies, calculated using the formula below, correspond to the total number of *Varroa* falls observed over the 10 weeks of Apivar^®^ treatment (Days 0 to 70 or D0–70) compared to the total number of *Varroa* mites fallen during the entire period (Apivar^®^ + Bayvarol^®^ and Oxybee^®^ control treatments; Days 0 to 112 or D0–112). 



## 6. Results

### 6.1. Laboratory Sensitivity of Varroa Mites Exposed to Amitraz LC_90_ and Genotyping of Varroa Mites in 2020

#### 6.1.1. Laboratory Sensitivity of *Varroa* Mites to the LC_90_ of Amitraz (2020)

Preliminary tests for the reference LC_90_ of amitraz in *Varroa* mites conducted in 2020 resulted in the determination of an LC_90_ of 28 ppm [CI95 (confidence interval 95): 22–36]. The coefficient of determination (R^2^) obtained was 0.8, indicating a good prediction of the linear regression. The correlation coefficient (r) was 0.9, indicating a high strength of the correlation between the variables “acaricide concentration” and “percentage of mite mortality”.

Out of the 18 colonies sampled for the main sensitivity test, valid amitraz sensitivity assay results were obtained for 751 *Varroa* mites (excluding negative controls) from 17 co-lonies across 9 different apiaries (Figure 2). The results from one colony were excluded because the mortality in the negative control group of mites, not exposed to amitraz, was too high (>30%, see Table 1). Of these 17 colonies, mite samples from 15 colonies were considered susceptible to amitraz, as per the pre-defined mortality rate of at least 75% (Table 1). *Varroa* mites from 2/3 of these colonies (*n* = 10) showed a mortality rate of 95% or higher during and after the exposure to amitraz LC_90_ in the laboratory. *Varroa* mite samples from three colonies demonstrated a mortality rate around the expected 90% mortality, between 89% and 94%. The remaining two mite samples that were classified as susceptible to amitraz demonstrated mortality rates of 77% and 78% (Table 1).

*Varroa* mites from two out of the seventeen sampled colonies were classified as showing intermediate sensitivity (60–75% mortality rate) in response to amitraz exposure in the laboratory (Table 1). None of the mite samples—and, consequently, the *Varroa* populations of the respective colonies—was classified as resistant against amitraz (Table 1). 

The overall LC_90_ mortality/paralysis rate for susceptible populations is 92% (*n* = 751), very close to the 90% that was expected, which supports the estimate of the LC_90_ obtained in preliminary tests.

No correlation was found in this study between the infestation rate and the level of sensitivity of *Varroa* mites. The four populations with the lowest mite sensitivity results (between 73% and 78% mortality rate) come from hives with *Varroa* brood infestation rate estimates between 1.9% and 5.4%. The 13 most sensitive mite populations (between 89% and 100% mortality rate) corresponded to the highly variable infestation rates in the respective honey bee colonies, with values between a minimum of 1% and a maximum value of 22.8% *Varroa* mite infestation.

#### 6.1.2. Genotyping of *Varroa* Mites Previously Tested in the 2020 Laboratory Assay

A total of 105 *Varroa* mites from all mites that were tested in the 2020 laboratory assays (53 and 52 mites, selected randomly among the dead and the surviving mites, respectively; see Table A1 and Table A2 for more details) were later genotyped and grouped into one of three genotypes, depending on the base pair detected at location 260 of their *ORβ-2R-L* gene: (1) A/A (wild-type) (*n* = 25), (2) A/G (heterozygous) (*n* = 15), and (3) G/G (homozygous variation) (*n* = 65). The genotyped mites originated from fifteen different colonies: twelve that were classified as sensitive and two that were classified as showing intermediate sensitivity to amitraz at the end of the laboratory sensitivity assay. The test did not provide valid results for the remaining colony (Table A1 and Table A2).

Thus, a total of 80 of the *Varroa* mites previously tested in the laboratory assay carried the variation at position 260 of the *ORβ-2R-L* gene, with 65 of them presenting a homozygous genotype.

Looking at the potential link between *Varroa* mite sensitivity to amitraz and the genotypes determined for 105 of the assay-tested mites, we were unable to identify a direct relationship between the variation at position 260 and a resistant phenotype in mites that had survived the assay. 

Table 2 indicates that, in both groups, mites that perished during or shortly after exposure to amitraz in the laboratory and mites that survived the assay, a majority of mites carried the previously identified variation. The percentage of mites carrying the variation was very similar in both groups (75% in mites that had died after amitraz exposure and 77.3% in mites that had survived amitraz exposure), not showing a significant difference (Chi2 = 0.0029765, df = 1, *p* = 0.9565) (Table 2). The only significant difference (Chi2 = 7.606, df = 2, *p* = 0.005817) between both groups (perished and surviving mites) is the distribution of homozygous and heterozygous individuals: 24.5% of surviving mites carried the A/G genotype, whereas the same was true for only 3.8% of perished mites (Table 2). Likewise, 52.8% of mites carried the homozygous G/G genotype in the group of surviving mites, compared to 71.2% among the perished mites.

### 6.2. Laboratory Sensitivity of Varroa Mites Exposed to Amitraz LC_90_, Treatment Efficacy of Apivar^®^ in the Field, and Genotyping of Varroa Mites in 2021

#### 6.2.1. Laboratory Sensitivity of *Varroa* Mites to the LC_90_ of Amitraz (2021)

To determine LC_90_, the results obtained for the two additional apiaries in 2021 were compiled with the results obtained for two other apiaries sampled in 2020, resulting in an estimate of the LC_90_ of 25 ppm [CI95: 21–29]. The coefficient of determination (R^2^) obtained was 0.7, which indicates a good prediction of linear regression, and the correlation coefficient (r) was 0.8, indicating a high correlation strength between the variables of acaricide concentration and percentage of mortality effect.

Although a total of 84 brood frames were sampled to test the laboratory sensitivity of *Varroa* mites to amitraz, only 25 in vitro assays resulted in valid sensitivity rates (8 samples from Occitanie and 17 samples from PACA). The other samples showed too-low infestation rates, and problems related to the excess mortality of *Varroa* mites in the brood during transport or in negative controls during the tests.

The average level of susceptibility of *Varroa* populations to LC_90_ (25 ppm amitraz) exposure is 76 ± 17%. This sensitivity varies from 58 ± 13% to 96 ± 6% depending on the beekeeping operation, with contrasting profiles depending on the region (87% ± 10% in Occitanie vs. 70 ± 17% in PACA) (Table 3). In total, of the 25 colonies sampled, 15 *Varroa* populations were classified as sensitive to amitraz (75–100% mortality), 6 were classified as intermediate (60–75% mortality), and 4 were classified as resistant (<60% mortality).

#### 6.2.2. Genotyping of *Varroa* Mites Previously Tested in the 2021 Laboratory Assay

A total of 135 *Varroa* mites (59 surviving mites and 76 dead mites at the end of the laboratory sensitive assay) from eight different colonies sampled for the laboratory sensitivity test and included in the field efficacy test with Apivar^®^ were genotyped in 2021. Two of the colonies had previously been classified as resistant in the laboratory sensitivity assay, five colonies were classified as showing intermediate sensitivity to amitraz, and the remaining one was classified as sensitive to amitraz (Table 4).

Similar to the genotyping results from the year 2020, the 2021 analyses revealed a higher percentage of mites carrying the homozygous G/G genotype compared to the homozygous wild-type A/A. This trend was observed in both groups of *Varroa* mites (those that survived and those that perished during amitraz exposure in the laboratory) (Table 5). Additionally, as in 2020, the heterozygous mites constituted the smallest portion of the three genotypes: 8.9% in 2021 vs. 14.3% in 2020 (Table 2 and Table 5).

Between the *Varroa* mites that survived amitraz exposure in the laboratory and *Varroa* mites which did not survive, there was only a slight difference (Chi2 = 2.5128, df = 1, *p* = 0.1129) in the percentage of mites carrying the variation (G/G) with a homozygous genotype (Table 5). When referring to the heterozygous genotype, as in 2020, surviving mites were more likely to carry this genotype, at 15.3% compared to 3.9% of heterozygous mites within the pool of mites that perished during the laboratory assay exposure to amitraz.

In a more detailed portrayal of the results, Figure 3 shows the partitioning of genotypes (A/A, G/G, and A/G) in individual mite samples (survivors vs. perished mites) for each individual colony and summarized across all genotyped mites (Figure 3).

Furthermore, Table A3 presents the individual results for each genotyped *Varroa* mite, detailing the origin, classification after the laboratory assay (sensitive, intermediate, or resistant), and the outcome of the assay (survived vs. perished).

Additional variations were found in the *ORβ-2R-L* gene (nature and positions of the variations described in the table reported in Table A4). We noted that most of them were detected in single surviving mites, making it difficult to establish an association with amitraz resistance in *Varroa* mites. The variated bases AT at positions 344–345 (found in 19 out of the 59 surviving mites), were present in surviving mites on the same allele as base A at position 260. 

#### 6.2.3. Treatment Efficacy of Apivar^®^ (2021) in the Occitanie and PACA Regions

Nine colonies were excluded from the final evaluation of the field efficacy: five in Occitanie and four in PACA. The reasons for exclusion included the need to replace the queen at the beginning of the test (*n* = 3), initial *Varroa* infestation being too low (<300 *Varroa* mites; *n* = 2), and significant weakening of the bee population at the end of treatment (“non-value” < 3000 bees; *n* = 4).

The mean efficacy in all tested colonies in both regions at the end of the 10-week treatment was 93% (SD ± 6.4) (*n* = 41 colonies). Twenty-one of these colonies demonstrated an efficacy of 95% or higher, seven of these in PACA and fourteen in Occitanie. Of the 20 colonies below this threshold, 9 did not achieve 90% efficiency (Figure 4). The mean field efficacy after 10 weeks in the two different regions was 95.5% (SD ± 2.7) in Occitanie (*n* = 14/20 > 95% Eff.; *n* = 2 < 90% Eff.) and 90.5% (SD ± 7.8) in PACA (*n* = 7/21 > 95% Eff.; *n* = 7 < 90% Eff.) (Table 6, Figure 5)

A comprehensive comparison of *Varroa* populations concerning amitraz sensitivity in the laboratory and Apivar^®^ treatment efficacy in the field across 50 colonies was not fully achieved. Despite sampling 84 brood frames, only 25 in vitro assays yielded valid *Varroa* mite mortality rates (8 samples from Occitanie and 17 from PACA). The remaining samples were compromised by low mite infestation, excess *Varroa* mortality in the brood during transport, or high mortality in negative controls during the tests.

The amitraz sensitivity–Apivar^®^ efficacy relationship could only be studied on 16 colonies (9 colonies were excluded from the field trial between *Varroa* mite sampling and the end of the trial). Despite the small amount of data, the results show a tendency towards better treatment efficacy in colonies with more sensitive *Varroa* populations (Spearman test, *p* < 0.05, rho = 0.5). This correlation is no longer significant when only the 11 colonies monitored in PACA are considered, revealing a regional effect. 

## 7. Discussion

### 7.1. Geographic Distribution and Spread and Magnitude of Amitraz Resistance Detected in the Field in French Apiaries

Over the course of two years (2020–2021), the laboratory sensitivity of *Varroa* mites to amitraz and the field efficacy of an amitraz-based *Varroa* treatment were investigated on mite populations of 75 honey bee colonies in 19 apiaries across France. The sampled apiaries were predominantly located in the Eastern, Central, and Southern regions of France. The main reasons for the focus on these geographic regions were as follows: (1) amitraz resistance in French *Varroa* mite populations was first reported from the Auvergne-Rhône-Alpes (AURA) region in the east of the country [26]; (2) the southern regions of France offer a high density of honey bee colonies and commercial beekeeping operations. 

The results of the laboratory assay in 2020, including *Varroa* mites from 17 colonies of 9 different apiaries, do not indicate widespread amitraz resistance in the sampled regions of AURA, Bretagne, Grand Est, and Occitanie. *Varroa* mite samples from 15 out of 17 sampled colonies were classified as “sensitive” to amitraz. However, it is important to note that, due to shipment constraints during the COVID-19 pandemic in 2020 and the resulting difficulties in testing *Varroa* mites for the laboratory assays when they were still alive, the total sample size of 17 colonies remains relatively low. On the other hand, the mites tested in the assay were sourced from nine apiaries across the country, including the AURA region in Eastern France and some apiaries in which resistance had previously been reported. For the investigation of the genetic basis of amitraz resistance in *Varroa destructor* (see below), the number of mites available for genotyping was more than adequate.

In 2021, we decided to focus on two regions in the South of France that are of high importance for the national honey production [40]: Occitanie and Provence-Alpes-Côte d’Azur (short: PACA). For this project, a field efficacy trial, consisting of a 10-week application with the authorized amitraz-based product Apivar^®^, was added to the labo-ratory assays. We found that, in both tests (laboratory assay and field efficacy trial), mites from the Occitanie region were largely sensitive to amitraz and the test results (mortality rate in the assay; treatment efficacy in percentage) were superior compared with the PACA region. With a treatment efficacy of 95.5% in the field, the result for the Occitanie region was compliant with the required minimum *Varroa* treatment efficacy of 95% for synthetic acaricides defined by the European Medicines Agency (EMA) [41]. The result of 90.5% efficacy in the PACA region was below the required minimum efficacy for synthetic treatments—the standard for organic *Varroa* treatments is slightly lower, with a required minimum efficacy of 90% [41]. 

The difference in field efficacy between the PACA and Occitanie regions may be explained by the regional differences in treatment history. Participating beekeepers from the Occitanie region reported that they had treated the honey bee colonies included in this trial with Apivar^®^ strips for 10–14 weeks in previous years (maximum treatment period of Apivar^®^ as per product label: 10 weeks). On the other hand, beekeepers from the PACA region reported that they had treated the colonies included in the present trial with Apivar^®^ for up to 24 weeks in previous years [42]. The prolonged and possibly repeated exposure of *Varroa* mite populations to an amitraz-based treatment which, in France, is authorized for a maximum treatment period of 10 weeks, could have contributed to the desensitization of *Varroa* mites in the PACA region to amitraz. For a more detailed analysis of the correlation between the off-label use of authorized and unauthorized amitraz-based *Varroa* treatments, we suggest further investigations including a higher sample size of honey bee colonies with a known treatment history. It is crucial for honey bee health, as well as the global beekeeping industry, to determine the potential long-term outcomes of the off-label use of *Varroa* treatments and raise awareness of the consequences of risking desensitization of *Varroa* mites against present and future treatment options.

Overall, the results of the present field efficacy trial support the hypotheses presented by Rinkevich (2020) [27]: 1. amitraz resistance occurs in “patches” or “islands” of resistance, rather than affecting all honey bee colonies located in a single apiary or even broader geographic regions within a short period of time; 2. the reduction in amitraz efficacy in the field clearly develops more slowly than pyrethroid or coumaphos resistance in *Varroa* mites [11,13,31]. 

In addition, this trial demonstrated that, across regions, the measured mortality rate of mites in the laboratory assay was lower compared to the corresponding field efficacy result (in percentage) after treatment with Apivar^®^. This indicates that mite sensitivity is likely not the only relevant factor for treatment efficacy in the field. This finding confirms similar results obtained in previous studies where field efficacy was compared to the mortality rate of *Varroa* mites in sensitivity assays [14]. A global result of 93% field efficacy (*n* = 41) corresponded to a laboratory mortality rate of 77.3% (*n* = 16). Even when we consider that the standard for treatment efficacy in the field is 95% mortality, and treatments such as Apivar^®^ have been formulated accordingly, when we exposed mites in the laboratory to the LC_90_, the mean difference between field efficacy and laboratory mortality was slightly higher than expected. This could be due to different factors, such as a higher natural mite mortality rate in honey bee colonies in the field [14] or a lack of long-term exposure of *Varroa* mites to amitraz in the laboratory, which does not correspond to the weeks-long treatment exposure in the field and therefore does not result in comparable mite mortality. While conditions in a live honey bee colony are clearly more variable than the standardized conditions in the laboratory, the mites have much more opportunity for avoidance behavior inside a honey bee colony compared to a spatially limited paraffin capsule in the laboratory. This contradicts the observed reduced laboratory mortality compared to the exposure in a honey bee colony. Future studies on *Varroa* mite resistance should consider the difference between laboratory mortality and field efficacy—which has been observed before—carefully. 

### 7.2. Genotyping of the ORβ-2R-L Gene and Potential Links to Amitraz Resistance

The genotyping analyses carried out in 2020 and 2021 included a total of 240 *Varroa* mites from 14 French apiaries in the eastern, central, and southern regions of France. The sequences constituted for all mites were compared to the sequence of the *ORβ-2R-L* gene referenced in NCBI.

The substitution of A with G at position 260 described by Hernández Rodríguez et al. [24] was identified, with a strong representativeness (70.8%) in all analyzed samples (70.8%), whatever their geographic localization and the status of their original colony, as defined by laboratory sensitivity tests (sensitive, intermediate, or resistant). Nevertheless, we were not able to establish a direct link between the presence of this variation and the development of a resistance to amitraz. Indeed, although many of the surviving mites of laboratory and field bioassays clearly carried the variation, no clear pattern in the distribution of the genotypes between perished and surviving mites was possible. Surviving mites did not carry the variation more often than mites that had died from amitraz exposure. Notably, out of the 170 mites carrying the variation at position 260, 143 (84.12%) were homozygous and 27 (15.88%) were heterozygous. As previously, it was difficult to establish a correlation between these two variated genotypes and a possible resistance to amitraz due to their similar distribution between the surviving and the dead mites.

Genotyping analyses allowed for the identification of other, previously undescribed variations in the *ORβ-2R-L* gene (Table A4). Two of them particularly drew our attention: the variations located at positions 344–345 that were detected in the *Varroa* mites analyzed in 2021 and were more frequently present in the “resistant” mites (ratio 19/3 between surviving and dead mites), suggesting a potential involvement in the acquisition of the resistance to amitraz. This hypothesis would merit further exploration. The involvement of these variations in the development of resistance would be inopportune given their low representativity (variations detected in 9.17% of the mites analyzed). 

The genome of *V. destructor* was, for a long time, believed to be particularly stable. Genetic bottleneck events, haplodiploid sex determination, and the mite’s sibling system contribute to this low genetic diversity [43]. However, in recent years, several studies have demonstrated that the genetic variation of the *Varroa* mite genome is higher than previously suspected, and that mite populations can differ significantly with respect to certain traits. *V. destructor* has as original obligate host the Asian honey bee, *A. cerana* (Hymenoptera; Apidae) [44]. In 1952, it switched from *A. cerana* to *A. mellifera* colonies in eastern Russia, producing what was named the Korean haplotype of the mite [45]. Another switch event occurred around 1957, leading to the creation of a second haplotype, the Japanese haplotype [45]. Both haplotypes (with a dominance of the Korean haplotype) first colonized the west of their countries of origin [46,47,48,49] and they are now present all around the world. These host switch events did not occur without any genetic changes. Several variants of both haplotypes were identified in different areas of Asia [49]. A more recent study suggests that the genomic diversity of the populations across the world could be even higher [50,51,52,53]. Whole genome analyses of 63 *V. destructor* and *V. jacobsoni* mites collected in their native ranges from both their ancestral and novel hosts allowed the identification of previously undiscovered mitochondrial lineages on the novel host, as well as the equivalent of tens of individuals involved in the initial host switch, demonstrating that a modest gene flow remains between mites adapted to their host [50]. Analyses of the fine scale population structure of *V. destructor* in a managed apiary setting highlighted the hierarchical genetic variation between apiaries, between colonies within an apiary, and even within colonies [54]. The authors also reported a modest increase in the total variation over time within individuals, possibly due to the de novo generation of diversity or, more probably, (due to the short time scales of the study) to the horizontal transmission of mites between colonies [54]. It is also now well known that a few populations of *A. mellifera* can survive *V. destructor* infestation without treatments by means of natural selection. A study was designed in 2016 to investigate potential genetic changes in mite populations from Sweden occurring in response to their host adaptation [55]. A comparison of mites collected in 2009 and 2017–2018 from *Varroa*-resistant *A. mellifera* populations and from neighboring mite-susceptible colonies highlighted significant changes in the genetic structure of the mites during the time frame of the study, with more pronounced variations in the *V. destructor* population collected in the mite-resistant honeybee colonies. This suggests that, like other parasites such as human lice [56], the human plasmodia, or mammal Leishmania [57], *V. destructor* reciprocally adapts to its host’s adaptations [55], and this coevolution occurs under the direct influence of environmental factors (climatic conditions, food resources, exposure to pesticides, and pollution). The variations that appeared in the genome of *V. destructor* following treatments with acaricides to develop resistance are perfect examples [24]. Another source of *V. destructor* genome variability is the possible hybridization with *V. jacobsoni* [58], another *Varroa* species mostly found within *A. cerana* but which causes less damage compared to *V. destructor*. With 99.7% of their genomes in common, both species are very similar genetically [59]. All events detailed above are all arguments that could explain the presence and the geographic specificity of the variation at position 260 (as well as the variations at positions 344–345), in addition to the development of a resistance to amitraz for which no correlation has been clearly established in this study. 

## 8. Conclusions

The present results call into question the hypothesis of a single-nucleotide polymorphism as the direct cause of amitraz resistance in *Varroa* mites, while an association between amitraz resistance and the occurrence of the variation at position 260 cannot be completely ruled out. However, we suspect that amitraz resistance development is more likely to include additional genetic variants and evolutionary steps, thus making the detection of amitraz resistance by markers more complex than detecting the presence of a SNP.

Further, the potential effects of off-label treatments (exposure to higher doses, a longer exposure period than recommended, the use of unauthorized treatments etc.) on resistance development should be the focus of future investigations. 

## Figures and Tables

**Figure 1 insects-15-00390-f001:**
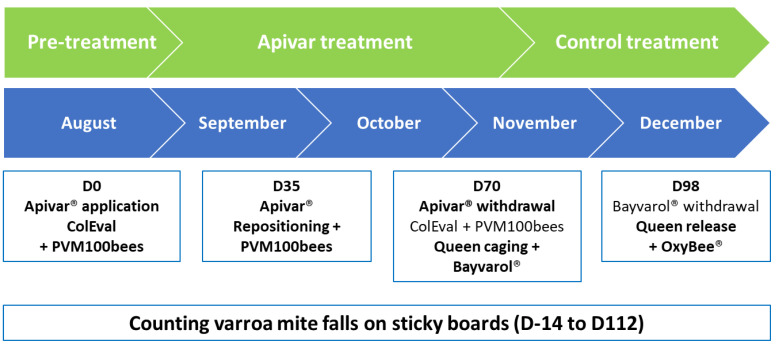
Time schedule of Apivar^®^ application in the field in the late summer of 2021.

**Figure 2 insects-15-00390-f002:**
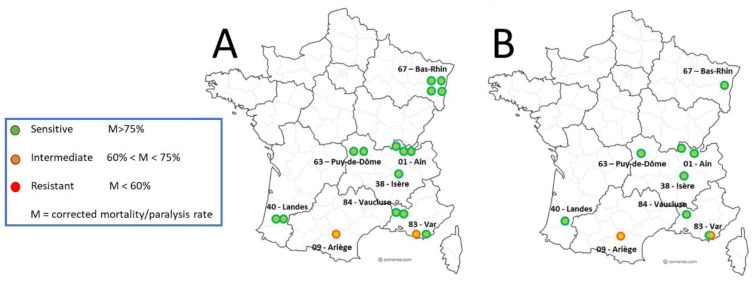
Laboratory sensitivity of *Varroa* mites exposed to amitraz LC_90_ and genotyping of *Varroa* mites; 2020 distribution. (**A**) Distribution map of *Varroa* population phenotyping—by colony (**A**) and by apiary (**B**).

**Figure 3 insects-15-00390-f003:**
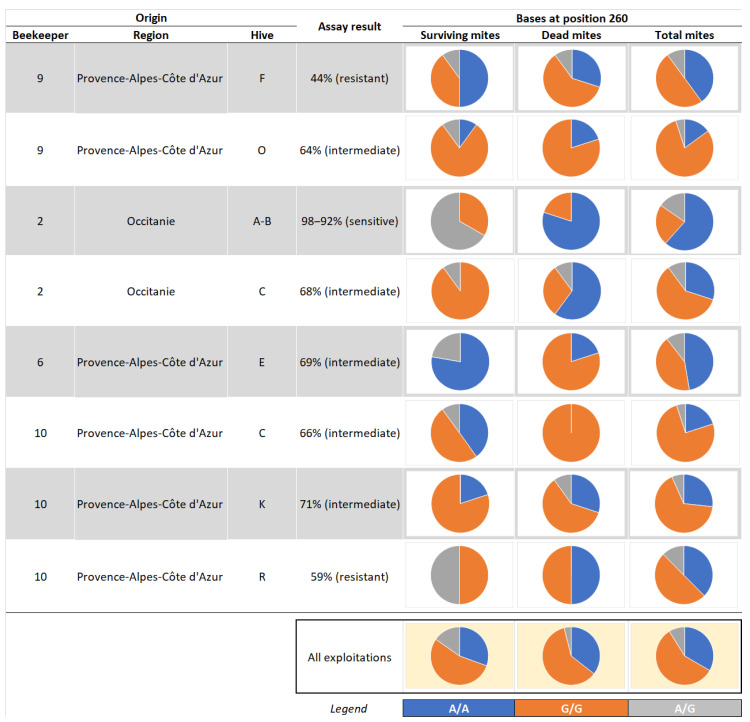
Partitioning of genotypes of *Varroa* mite samples collected in 2021 after mites’ exposure to the LC_90_ of amitraz in the laboratory.

**Figure 4 insects-15-00390-f004:**
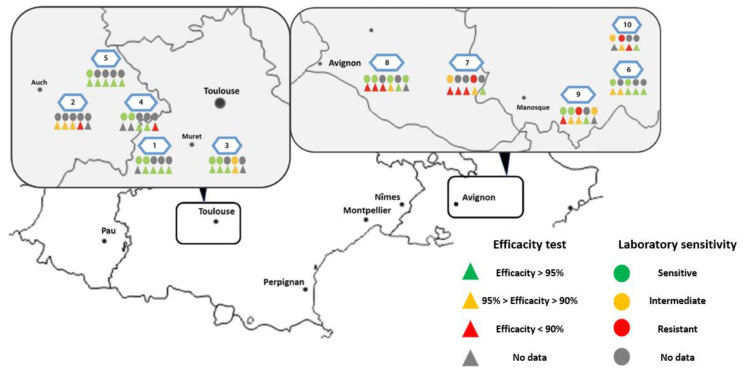
Map presenting field efficacy and sensitivity results assessed per colony in each apiary.

**Figure 5 insects-15-00390-f005:**
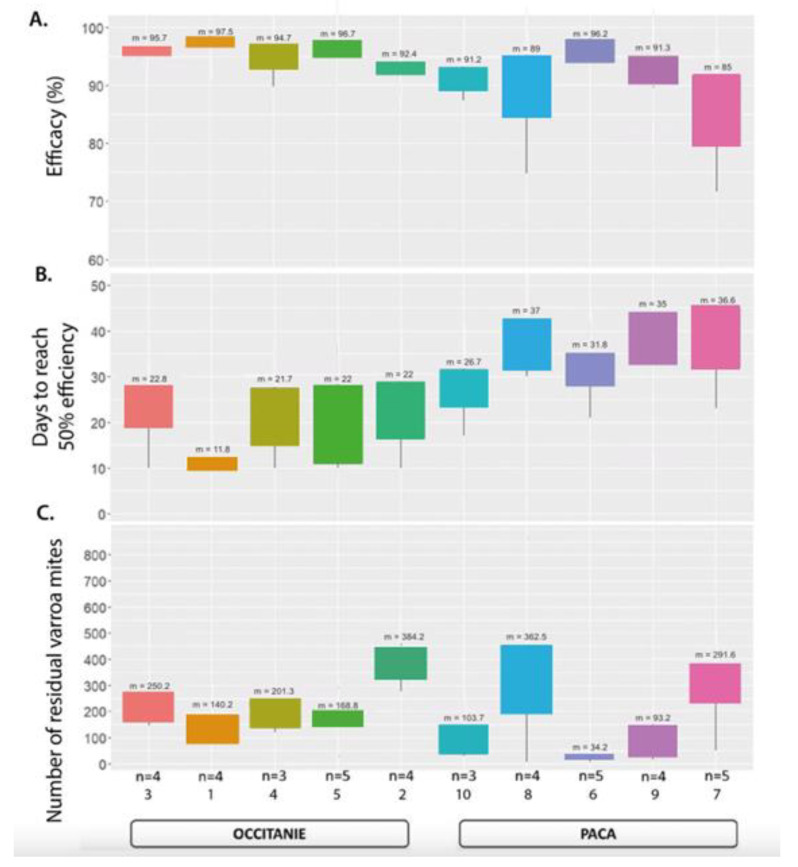
Boxplots of calculated efficacy for the Apivar treatment (**A**), days to achieve 50% efficacy (**B**), and numbers of residual *Varroa* mites after treatment (**C**). Numbers (*n*) of hive per apiary and means (m) are indicated.

**Table 1 insects-15-00390-t001:** Overview of the mortality rate of *Varroa* mites after exposure to the LC_90_ of amitraz in the laboratory (2020). The sample size (*n*) represents the total number of mites tested over six French regions: Nouvelle-Aquitaine, Occitanie, Provence-Alpes-Côte d’Azur, Auvergne-Rhône-Alpes, Centre-Val de Loire, and Grand Est. The mortality rates of those total samples are given in percentage (%) for LC_90_ test assays and negative controls.

Origin		Dead or Paralyzed Mites Rate in Percentage (%) at Day 0 + 24 h			Classification
Apiary	Hive	LC_90_ (n)	Negative Control (n)	Corrected LC_90_ (n)	*Varroa* population
1	A	90% (60)	13% (31)	0.89	sensitive
1	B	78% (60)	3% (30)	0.78	sensitive
3	C	74% (50)	3% (30)	0.73	intermediate
4	D	95% (60)	2% (43)	0.95	sensitive
4	E	98% (60)	3% (40)	0.98	sensitive
5	F	95% (59)	3% (38)	0.95	sensitive
5	G	79% (14)	17% (6)	0.74	intermediate
6	H	77% (56)	0% (30)	0.77	sensitive
7	I	100% (13)	0% (10)	1	sensitive
7	J	94% (32)	0% (10)	0.94	sensitive
8	K	91% (90)	0% (38)	0.91	sensitive
9	L	100% (20)	0% (10)	1	sensitive
9	M	96% (25)	0% (10)	0.96	sensitive
9	N	100% (15)	0% (7)	1	sensitive
9	O	96% (46)	0% (20)	0.96	
10	P	100% (21)	38% (8)	non valid (Centre-Val de Loire región)	
11	Q	98% (42)	7% (15)	0.97	sensitive
11	R	98% (49)	0% (20)	0.98	sensitive

**Table 2 insects-15-00390-t002:** Genotyping results for *Varroa* mites, sampled after LC_90_ exposure to amitraz. Results show the base at position 260 of the *Varroa* mite *ORβ-2R-L* gene (2020).

Mites Sampled after LC-90 Exposure Towards Amitraz	N (%)		
	Base at Position 260 in the ORβ-2R-L		
	A/A	G/G	A/G
Perished mites	13 (25%)	37 (71.2%)	2 (3.8%)
Surviving mites	12 (22.6%)	28 (52.8%)	13 (24.5%)
**Total mites**	**25 (23.8%)**	**65 (61.9%)**	**15 (14.3%)**

**Table 3 insects-15-00390-t003:** Overview of the mortality rate of *Varroa* mites after exposure to the LC_90_ of amitraz in the laboratory (2021). The sample size (*n*) represents the total number of mites tested. Mortality rates of those total samples are given in percentage (%) for LC_90_ test assays and negative controls. (Géographic maps.)

Origin			Dead or Paralyzed Mites Rate in percentage (%) at Day 0 + 24h			Classification
Region	Beekeeper	Colony	LC_90_ (n)	Negative Control (n)	Corrected LC_90_ (n)	*Varroa* population
Occitanie	1	1.1	93% (42)	0% (10)	0.93	sensitive
		1.5	87% (46)	0% (10)	0.87	sensitive
	2	2.3	92% (60)	40% (40)	non-valid	
		2.4	93% (56)	40% (30)	non-valid	
	3	3.1	73% (70)	15% (20)	0.68	intermediate
		3.2	98% (59)	0% (30)	0.98	sensitive
		3.3	94% (49)	23% (22)	0.92	sensitive
	4	4.5	81% (16)	0% (6)	0.81	sensitive
		4.6	85% (176)	0% (60)	0.85	sensitive
	5	5.1	96% (28)	25% (8)	0.95	sensitive
**Total (Occitanie)**					**87% (± 10)**	
PACA	6	6.3	91% (22)	0% (8)	0.91	sensitive
		6.5	100% (20)	10% (10)	1	sensitive
	7	7.1.3	66% (70)	0% (24)	0.66	intermediate
		7.1.11	71% (35)	0% (10)	0.71	intermediate
		7.1.18	60% (49)	0% (19)	0.60	intermediate
		7.2.3	64% (14)	13% (8)	0.59	resistant
		7.2.5	40% (77)	6% (50)	0.36	resistant
	8	8.2	75% (20)	0% (3)	0.75	sensitive
		8.3	83% (81)	0% (49)	0.83	sensitive
		8.6	89% (80)	3% (39)	0.88	sensitive
		8.8	86% (80)	2% (50)	0.86	sensitive
	9	9.15	64% (80)	0% (50)	0.64	intermediate
		9.6	44% (80)	0% (50)	0.44	resistant
		9.1	79% (14)	0% (4)	0.79	sensitive
		9.3	65% (16)	0% (7)	0.75	sensitive
	10	10.5	69% (29)	0% (8)	0.69	intermediate
		10.7	48% (59)	0% (40)	0.48	resistant
**Total (PACA)**					**70% (± 17)**	
**Total (Global)**					**76% (± 17)**	

**Table 4 insects-15-00390-t004:** Overview of genotyped *Varroa* mites from laboratory assay samples, presenting the mites´ origin, laboratory sensitivity results of the colony, and the status of the mites after amitraz exposure in the laboratory (survivors vs. dead *Varroa* mites).

Origin			Assay Result	Survivors Mites Available	Dead Mites Available	Proposed Genotyping	Total No of Genotyped Mites
Beekeeper	Region	Hive					
9	Provence-Alpes-Côte d'Azur	F	44% (resistant)	39	40	10 vs. 10	20
9	Provence-Alpes-Côte d'Azur	O	64% (intermediate)	24	45	10 vs.10	20
2	Occitanie	A	98% (sensitive)	1	9	3 vs. 10	13
2	Occitanie	B	92% (sensitive)	2	17		
2	Occitanie	C	68% (intermediate)	11	39	10 vs. 10	20
6	Provence-Alpes-Côte d'Azur	E	69% (intermediate)	9	20	9 vs. 10	19
10	Provence-Alpes-Côte d'Azur	C	66% (intermediate)	22	38	10 vs. 10	20
10	Provence-Alpes-Côte d'Azur	K	71% (intermediate)	5	15	5 vs. 10	15
10	Provence-Alpes-Côte d'Azur	R	59% (resistant)	2	6	2 vs. 6	8

**Table 5 insects-15-00390-t005:** Genotyping results for *Varroa* mites, sampled after LC_90_ exposure to amitraz. Results show the base at position 260 of the *Varroa* mite *ORβ-2R-L* gene (2021).

Mites Sampled after LC-90 Exposure towards Amitraz	N (%)		
	Bases at Position 260 in the ORβ-2R-L		
	A/A	G/G	A/G
Surviving mites	18 (30.5%)	32 (54.2%)	9 (15.3%)
Dead mites	27 (35.5%)	46 (60.5%)	3 (3.9%)
**Total mites**	**45 (33.3%)**	**78 (57.8%)**	**12 (8.9%)**

**Table 6 insects-15-00390-t006:** Summary table of amitraz field efficacy results (SD averages, both globally and segmented by region).

	Occitanie (n = 20)	PACA (n = 21)	Global
Efficacy ± SD (%)	95.5% ± 2.7	90.5% ± 7.8	93% ± 6.4
Colonies >95% efficacy (n)	14	7	21
Colonies [90–95%] efficacy (n)	4	7	11
Colonies <90% efficacy (n)	2	7	9

## Data Availability

The authors confirm that the data supporting the findings of this study are available within the article and/or its.

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
