# Peer review of "Amitraz Resistance in French Varroa Mite Populations—More Complex Than a Single-Nucleotide Polymorphism"

_insects, 2024, doi:10.3390/insects15060390_

Round 1
Reviewer 1 Report
Comments and Suggestions for Authors
The paper requires major corrections because it is not written in a form that is adequate for a scientific paper.
Apart from the necessary proofreading (because of syntax errors, a lot of redundant words ...), the structure of the work must also be corrected because it is too extensive, contains are too many repetitions ...). A scientific paper must be written concisely.
Below I give some suggestions for improving the work. The same comments are included in the attached PDF:
Line 38: Add recent reference related to colony losses:
· Bruckner S, Wilson M, Aurell D, Rennich K, Vanengelsdorp D, Steinhauer N, Williams GR. (2023) A national survey of managed honey bee colony losses in the USA: results from the Bee Informed Partnership for 2017–18, 2018–19, and 2019–20. Journal of Apicultural Research, 62(3):429-443.
- Lines 48-53: Please shorten this (well-known) historical data.
- Lines 56-58: I suggest you to shorten this sentence as follows: "Once it occurred, the resistance of Varroa mites against pyrethroids spread quickly and led to severe reductions in treatment efficacy."
- Lines 60-67: Please consider mentioning these references:
· Alissandrakiset et al: Pyrethroid target site resistance in Greek populations of the honey bee parasite Varroa destructor (Acari: Varroidae). Journal of Apicultural Research, 2017; 56(5):625-630.
· Erdem et al. Geographical distribution of pyrethroid resistance mutations in Varroa destructor across Türkiye and a European overview. Experimental and Applied Acarology, 2024; 92:309–321.
· Yarsan et al. Investigation of resistance against to flumethrin using against Varroa destructor in Türkiye. Veterinary Research Communications, 2024; https://doi.org/10.1007/s11259-024-10351-x
- Lines 69-71: Please correct this sentence in accordance with the recently published paper of Brodschneider et al. (2023) who found that:
- Most colonies (62%) in Europe are treated with amitraz (in strips or fumigation).
- In eight Western European countries (cluster I) amitraz strips are dominantly used.
- In seven Eastern European countries (cluster III) amitraz fumigation is dominantly used.
The paper Brodschneider et al. (2023) is available at: https://link.springer.com/article/10.1007/s10340-022-01523-2
· Brodschneider et al. Spatial clusters of Varroa destructor control strategies in Europe. Journal of Pest Science 2023; 96(2):759-783.
- Lines 90-91: „potential manifestation“ – Do you mean „potential cause“?
- Lines 104-105: „It results in a tyrosine (TAT) to histidine (CAT) substitution“ should read „tyrosine to histidine substitution due to the transition in the corresponding code (TAT to CAT).“
- Line 109: I suggest you to include references I already listed above:
· Alissandrakis et al. (2017)
· Erdem et al. (2024)
· Yarsan et al. (2024)
- Lines 115-125: This part should be shortened and corrected. This is a crucial part of the paper, so it must be clear. Please be concise. First, list the goals, and then the activities you have taken to achieve those goals. Nothing else should be written here.
- Line 144: „2.2.1. Determination of Reference LC90 Concentration of Amitraz“ - This whole subsection should be rewritten - prepared in a way adequate for a scientific paper (The way of writing is more like for a dissertation than for a paper in a scientific journal).
- Line 153: Please introduce the abbreviation here as follows: Laboratoire Départemental d'Analyses du Jura (LDA39).
- Line 170: „capsule“ - What capsules?
- Line 175: „the method adopted comes ...“ - What method?
- Line 177: See the comment in Lines 796-797: related to Ref 30.
- Line 193: „No. 277 • 1-2 / 2017“
- Line 216: „in the same title“ - inadequate English translation.
- Lines 220-229: Please correct this to be adequate for scientific paper.
- Lines 232-233: I suggested introducing the abbreviation "LDA39" in Line 153 (at first appearance) so here only "LDA39" should be left.
- Lines 255-261: Please write this correctly so it can be understood.
- Lines 264-266: This is not properly written. Is it formula?
- Line 279: „% effect“ - Please write this correctly so it can be understood.
- Line 289: „UP“ - What does it mean (ultrapure?).
- Lines 320-324: If you are not the first to implement this in the manner described, please refer to the reference where it was described earlier.
- Line 320: „floor“ should be replaced with „bottom-board“.
- Line 321: placed within mesh.
- Line 341 (top of page 8): There is an inserted unnamed formula that is not mentioned in the text.
- Lines 346-347: „The results .... was carried out" - Something is wrong here, please check and correct it.
- Lines 347-348: This belongs to the Material & Method section.
- Lines 400-401: „Hernández Rodríguez et al. [24]“ - Please delete. This is repeatedly mentioned in the Introduction. It should be mentioned in "Material and Methods", but not in the "Results".
- Line 408: „Hernández Rodríguez et al. [24]“ - Please delete. It does not belong to the "Results".
- Lines 464: „(the variation, detected by Hernández Rodríguez et al.)“. Please delete. It does not belong to the "Results".
- Lines 493-494: „suggesting an involvement in the acquisition of the resistance to amitraz. This hypothesis would merit further exploration“ - This belongs to the "Discussion" section.
- Figure 4: The title is incomplete. Whose efficiency? Whose sensitivity?
- Table 6: The title is incomplete. Whose efficiency?
- Lines 528-529: „This indicates mite sensitivity is likely not the only factor relevant for treatment efficacy in the field.“ - This belongs to the „Discussion“ section.
- Lines 533-536: This is very important sentence but it is too broad and therefore difficult to understand. Please make adequate corrections.
- Line 582: „2. Treatment efficacy reduction“ - Please replace with "The reduction of amitraz efficacy".
- Line 584: „Further, it might be closely related“ - Please define „it“.
- Line 594: „mean difference is slightly higher“ - Please clarify the "difference".
- Lines 643-649: In this part you should also refer to following references (and new Varroa destructor haplotypes):
· Gajic et al. Variability of the honey bee mite Varroa destructor in Serbia, based on mtDNA analysis. Experimental and Applied Acarology 2013; 61: 97-105.
https://link.springer.com/article/10.1007/s10493-013-9683-9
· Gajić et al. Haplotype identification and detection of mitochondrial DNA heteroplasmy in Varroa destructor mites using ARMS and PCR–RFLP methods. Experimental and Applied Acarology, 2016; 70: 287-297.
https://link.springer.com/article/10.1007/s10493-016-0086-6
· Gajić et al. Coexistence of genetically different Varroa destructor in Apis mellifera colonies. Experimental and Applied Acarology. 2019; 78:315-326.
https://link.springer.com/article/10.1007/s10493-019-00395-z
- Lines 680-681: „additional steps and perhaps other, interacting variations“ - Please improve to be clear what you want to say. Clearly state the conclusions drawn from your results.
- Lines 683-688: This part belongs to the "Discussion" section.
REFERENCES:
- In 13 references, the authors are not properly written (highlighted in the attached PDF).
- Lines 745-746: be careful with the results published only in the preprint version. For ref. 8 there is a note: „This article is a preprint and has not been certified by peer review“, see: https://www.biorxiv.org/content/10.1101/2023.03.22.533871v1
- Line 786: Are you sure that reference No. 25 is a relevant source of information, especially from a scientific point of view? Besides, there is no Vol and no pagination in this reference.
- Lines 796-797: Ref 30. Please write the reference properly. Here is the citing example of an article from the BEEBOOK:
· Dietemann, V., Nazzi, F., Martin, S. J., et al. (2013). Standard methods for Varroa research. In V. Dietemann, J. D. Ellis, & P. Neumann (Eds.), The COLOSS BEEBOOK, Volume II: Standard methods for Apis mellifera pest and pathogen research. Journal of Apicultural Research, 52(1), 1–54. https://doi.org/10.3896/IBRA.1.52.1.09
- Figures 1, 2, 4 and 5 are of poor resolution.
- Table 3 and Appendices (I, II, III and IV) should be submitted as Supplementary Material

Comprehensive corrections related to English are necessary.
Author Response
Dear reviewer,
Please find attached the answers to your comments.
Best regards,
Abderrahim.

Reviewer 2 Report
Comments and Suggestions for Authors
I consider that the manuscript is very good, it reports valuable information at a global level on the control of the most important disease for honey bees. I have no relevant suggestions for changes, only details such as: improving the quality of the images (figure 1 and 2), homogenizing the use of the word varroa, Varroa, I think it should be in italics and in capital letters because it is a genus, replacing the percentage symbol in the text by the word percentage. On line 355 it does not have a point.
Author Response

(The authors gave the same response as above.)

Reviewer 3 Report
Comments and Suggestions for Authors
the topic of this manuscript is very interesting.
Showing that a chemical compound is highly active against varroa and that there are varroa susceptible and resistant to the substance by linking them to a molecular mechanism is a good achievement for research in this field.
It would have been interesting during field monitoring of varroa mortality at treatment to see bee mortality in parallel.
The article is well designed scientifically and the results are clear. I suggest implementing the conclusions which are a quite general.
Minor comments:
-page 2 lines 46-48 Bibliographic references are missing. Please add them.
-page 2 line 80 Which assays are the labs referring to? please implement this part.
-page 5 line 212 Please specify why 1 hour was chosen as the time.
-page 5-6 I suggest adding a table to clarify the sampling times because it is not very clear this way.
-page 6 line 265 Is the fraction sign missing in this formula? if so please add it.
-page 7 line 305 I think "They" is a typo. Please remove it.
-pages 9-10 lines 401 and 408 generally do not put bibliographic references in the results. I suggest specifying this reference in the materials and methods.
-Tables 3, 4 e appendici I-IV the lettering is too small, reads poorly or not at all. Please enlarge the fonts.
-page 13 line 487 “vs” should be written in italics. Please correct it.
-page 13 lines 492-494 this part belongs to the discussion section. Please move it.
-page 14 line 519 “in vitro” should be written in italics. Please correct it.
-page15 lines 528-529 this part belongs to the discussion section. Please move it.
-page 17 lines 633-637, 642, 655-657 Bibliographic references are missing. Please add them.
- I recommend to implement the conclusion section.
Author Response

(The authors gave the same response as above.)

Round 2
Reviewer 1 Report
Comments and Suggestions for Authors
The authors corrected the manuscript following my suggestions. However, two aspects have remained disputed: the technical aspect (way of formatting the work) e.g. see Lines 220-228, Lines 246-269, Lines 292-297; errors in writing (mainly syntax errors).
I also noticed some mistakes that need to be corrected (details below):
Line 62: first letter in word „Resistance“ should be small.
Line 121: Include ref. 25. next to 20, 21, 22.
Line: 135: Do you mean: „Assessment of treatment efficacy of an authorized Varroa control product“?
Line 136: Do you mean „Assessment of laboratory sensitivity of Varroa mites towards amitraz“?
Line 141: Do you mean „Design of the study“?
Line 189: Please check the reference number: I think you mistakenly wrote 23, instead of 32 (I conclude this based on the following: (1) ref. 32 is skipped, and (2) the sense of the citation indicates that ref. 23 cannot be cited there.
Line 193: Why did you write „Milani et al.,“ and refere to the ref. No. 35 written only by Milani (without coauthors)?
Line 212: You wrote Milani et al. without the reference No. You have reference 32, written by Milani and Vedova, 2022, and Milani (1995).
Lines 602-603: It should be „V. destructor“ instead of Varroa destructor, „A. cerana“ instead of „Apis cerana“.
Line 611-612: Include references 52 and 53 next to 50, 51. Besides „carried out by the authors“ should be removed.
Lines: 616: Only reference 51 should be left here (because whole genome analyses were not done in references 51, 52, and 53).
Line 622: It should be „V. destructor“ instead of Varroa destructor.
Line 636: It should be „V. destructor“ instead of Varroa destructor.
Line 793: Correct the surname of the first author (it should be Milani).
Line 803: Correct the surname of the author (it should be Milani).
REFERENCES:
The references are carelessly written, especially the newly added ones (they are not uniform and the instructions of the journal are not respected).
Comments on the Quality of English LanguagePartially improved, still requires editing.
Author Response
We appreciate all the comments received and proceed to show the list of corrections made
